# Corrections of Molecular Morphology and Hydrogen Bond for Improved Crystal Density Prediction

**DOI:** 10.3390/molecules25010161

**Published:** 2019-12-31

**Authors:** Linyuan Wang, Miao Zhang, Jie Chen, Liang Su, Shicao Zhao, Chaoyang Zhang, Jian Liu, Chunyan Chen

**Affiliations:** 1School of Chemistry and Chemical Engineering, Southwest Petroleum University, Chengdu 610500, China; 13981884015@163.com (L.W.); zmhmzf@163.com (M.Z.); cychen986@163.com (C.C.); 2Institute of Chemical Materials, China Academy of Engineering Physics (CAEP), P.O. Box 919-311, Mianyang 621999, China; chaoyangzhang@caep.cn; 3College of Computer Science and Technology, Southwest University of Science & Technology, Mianyang 621010, China; rokiecoder@163.com (J.C.); hurrysu@163.com (L.S.); 4Institute of Computer Applications, China Academy of Engineering Physics (CAEP), P.O. Box 919-1201, Mianyang 621999, China; zhaoshicao@gmail.com

**Keywords:** DFT, density prediction, QSPR

## Abstract

Density prediction is of great significance for molecular design of energetic materials, since detonation velocity linearly with density and detonation pressure increases with the density squared. However, the accuracy and generalization of former reported prediction models need further improvement, because most of them are derived from small data sets and few molecular descriptors. As shown in this paper, for molecules presenting brick-like shape or containing more hydrogen-bond donors the predicted densities have large negative deviations from experimental values. Thus, a molecular morphology descriptor *η* and a hydrogen-bond descriptor *H_b_* are introduced as correction items to build 3 new QSPR models. Besides, 3694 nitro compounds are adopted as data set by this work. The accuracies are obviously improved, and the generalizations are verified by an independent test set. At the level of B3PW91/6-31G(d,p), the effective ratios (ERs) of the 3 Equations, for *Δρ* < 5%, are 92.7%, 91.8%, and 93.3%; for *Δρ* < 2%, the values are 53.5%, 51.3%, and 54.7%. At the level of B3LYP/6-31G**, for *Δρ* < 5%, the values are 92.3%, 91.4% and 92.9%; for *Δρ* < 2%, the values are 53.7%, 51.4% and 53.2%.

## 1. Introduction

Crystal density is an intrinsic characteristic for solids, which is used as an index to reveal material properties including mechanistic, thermodynamic, etc. [1,2,3,4,5,6,7,8,9,10,11,12]. As computational chemistry and computational physics are becoming increasingly indispensable in the field of material design, accurate density prediction is of great significance for materials with density-sensitive properties. For instance, to predict a proper detonation performance of an energetic material, it would be necessary to have a reliable estimation of density while considering that the velocity of detonation linearly with density and the detonation pressure increases with the density squared [13].

Since reliable density prediction is the utmost important for estimation of detonation performances, various approaches have been developed to predict densities of energetic compounds over the decades [14,15,16,17,18,19,20,21,22,23,24]. In general, methods for density prediction of organic crystals can be summarized as (i) to obtain the density of single crystal via crystal structure prediction (CSP), (ii) directly calculate via empirical formula based on “group addition” methods (GAM), (iii) combined methods of density functional theory (DFT) and quantitative structure-property relationship (QSPR). In the third category, DFT calculations are applied to obtain chemical descriptors for QSPR.

In all of the above methods, CSP is the most reliable one to calculate crystal density. CSP aims at predicting an unknown crystal structure based on a given molecular structure. The procedures of CSP include random sampling, structure optimization, and energy ranking [25] in which the accuracy of energy calculation is the decisive factor [26]. CSP has been proved to be effective in predicting the structure of inorganic or metallic crystals, but it is not good at dealing with crystals composed of organic molecules [13,27,28,29]. For one thing, conformations and orientations of molecular crystals involving more structural freedom than inorganic or metallic crystals, even powerful supercomputers struggle to provide enough computing resources to achieve effective predictions. For another, the calculation of the weak interaction energy between molecules is a most formidable task, whether through first principle or molecular mechanics, unfortunately The packing patterns of molecules are greatly influenced by these intermolecular interactions [30]. Using CSP to predict the density of organic molecules remains a difficult task, since predicting crystal structures of these materials remains a challenging work [31].

The easiest method to perform density prediction is GAM, which requires only a set of group volumes that can be summed to estimate an effective molecular volume through empirical and simple theoretical function [14,15,24,32,33,34,35,36]. GAM can obtain the effective volume and density of molecules in a relatively short time, its inherent disadvantage is but the low accuracy. In most of the GAM expressions, the descriptors, such as molecular configuration, conformation, and non-bond interaction that have decisive influences on packing pattern of molecules, are missing, and the density prediction errors affected by them are difficult to be corrected through simple empirical formulas. Besides, GAM requires the manual identification of substructures due to the detailed classification of chemical groups and substructures, thus it is not capable of automatic machine predicting when large quantities of compounds need to be marked. [25] In other words, the intelligent algorithms for molecular disassembly must be available before GAM being applied to computer programs, but to achieve these algorithms has been proved to be a challengeable work [31,37,38,39,40,41,42].

DFT + QSPR is the most widely accepted method for density prediction, by which the molecular volume V_m_ can be readily derived from contour of the molecular electronic density (CMED) that obtained via DFT calculation, and then it will be extrapolated to the crystal density via a given QSPR model. [14,15] The most common used value of electronic density in drawing CMED is 0.001 electrons/bohr^3^ [17,18,19,20,43,44,45,46,47,48]. Politzer et al. calculated the CMED of 36 neutral molecules at the calculation level of B3PW91/6-31G(d,p), and they introduced positive and negative charge separation as a correction item in their QSPR model, in which the prediction error of 78% compounds was within 0.050 g/cm^3^. [21] Rice et al. fitted a QSPR model using 180 neutral and 23 ionic molecules at the calculation level of B3LYP/6-31G**, in which the root mean square (RMS) percent deviation and average absolute error of density prediction are 2.7% and 0.035 g/cm^3^, respectively [23].

Although the density prediction method of DFT + QSPR has been widely used in molecule design and performance prediction, its accuracy needs further improvement. For most of the reported DFT + QSPR methods, the training set is less than 300 samples, with very few molecular descriptors. These hindrances will affect the training accuracy and generalization of the QSPR models. Current work intends to improve density prediction models of DFT + QSPR to perform better accuracy and generalization than before, by introduces new molecular descriptors and extends training set to thousands of samples. As is known to all, the density of organic crystals is not only determined by molecular volume and charge, but also influenced by morphology [19,49], hydrogen bond [50], π-π packing [51], and other factors. The effects of molecular morphology and hydrogen bond on crystal density were not considered by the reported DFT + QSPR methods, which is likely to have an impact on the accuracy of the density prediction. 

To improve the accuracy of the model in density prediction of energetic compounds the effects of molecular morphology and hydrogen bond were quantitatively described and introduced into the DFT + QSPR models. The training set and testing set used in this study were 3694 nitro compounds extracted from Cambridge Crystallographic Data Centre (CCDC), among which the training set and testing set accounted for 50%, respectively. In order to quantitatively describe the molecular morphology and the effect of hydrogen bond, a molecular morphology descriptor *η* and a hydrogen bond descriptor *H_b_* were defined. The results show that *η* and *H_b_* were significantly correlated with the negative deviation of predicted densities. For molecules presenting brick-like shape or containing more hydrogen-bond donors, the predicted densities tend to be lower than the experimental values. Therefore, the two new molecular descriptors, *η* and *H_b_*, were introduced into the former formulas [21,22,23] as correction terms (see Section 3.2 and Section 3.3). 3 QSPR models were proposed by using both the descriptors separately and together. The former reported two calculation levels B3PW91/6-31G(d,p) and B3LYP/6-31G** were applied, both considering D3 correction. The models and the levels were mixed to 6 working conditions. For all the working conditions fitted by this work, the computational accuracy was greatly improved compared with all the previous reports of DFT + QSPR. The prediction accuracy of the testing set agrees perfectly with that of the training set, to prove that the models of the work have good generalization. 

## 2. Results and Discussion

### 2.1. Corrections Ananysis on Prediction Error

To evaluate the influence of *η* and *H_b_* on prediction error, the relationships between each descriptor and the prediction error are analyzed (Figure 1), using relative error *Δρ* between the predicted density *ρ_cal_* via Equation (11) [47,48] and the experimental density *ρ_exp_*.

The negative deviation of *Δρ* increasing with the increase of *η* and *H_b_*, indicating that density values of the molecules with greater *η* or *H_b_* are more underestimated. It has been reported that planarity [21] and hydrogen-bond [50] are the most important two factors beneficial to dense packing of crystal. Therefore, Equations, without considering these two factors will result in lower density prediction values, especially for molecules with more ‘bricklike’ shape or more hydrogen bonds. Adding *η* and *H_b_* as correction items in the conventional QSPR Equations is expected to improve the accuracy of density prediction, which is performed in subsequent fitting works.

### 2.2. Construction of Correction Formulas

The fitting works are performed according to Equations (1)–(3).
(1)$$\rho =\alpha \left(\frac{M}{V_{m}}\right)+\beta \left(\nu \sigma_{tot}^{2}\right)+\beta_{1}\eta +\gamma$$
(2)$$\rho =\alpha \left(\frac{M}{V_{m}}\right)+\beta \left(\nu \sigma_{tot}^{2}\right)+\beta_{2}H_{b}+\gamma$$
(3)$$\rho =\alpha \left(\frac{M}{V_{m}}\right)+\beta \left(\nu \sigma_{tot}^{2}\right)+\beta_{1}\eta +\beta_{2}\left(\frac{H_{a}}{M}\right)+\gamma$$

For the case that molecular morphology alone influences density prediction, *η* is added as a correction term in Equation (1). For the case that hydrogen bond alone influences density prediction, *H_b_* is added as a correction term in Equation (2). Considering the coupling influences from molecular morphology and hydrogen bond, where both *η* and *H_b_* are added as correction terms in Equation (3). The correction items $\nu \sigma_{tot}^{2}$, *η* and *H_b_* have enough physical meanings, so the system correction factor α can be shielded, and all the values of α are set to 1 in the following fitting works.

### 2.3. Fitting Results

According to the 3 Equations and 2 calculation levels in Section 2.2, the following 6 working conditions can be obtained to perform fitting works (Table 1).

The fitting works are performed using Orthogonal Distance Regression [52], and the results are illustrated in Figure 2, by comparing the calculated density *ρ_exp_* with the experimental density *ρ_exp_*.

As shown in Figure 2 that the density values that were obtained from QSPR, for all the 6 working conditions, are in good agreement with that of experiments. Overall, the values of MSE are less than (12), indicating that the averaged fitting error of *Δρ*, which can be easily evaluated from quadratic root of the MSE, for each of the 6 conditions is less than 3%. The orders of MSI(III) < MSI(I) < MSI(II) and MSI(VI) < MSI(IV) < MSI(V) indicate that the training accuracy of the 3 Equations are ranked as (3), (1) and (2). Equation (3), corrected by both *η* and *H_b_*, presents the best training accuracy, while the training accuracy of Equation (1) with correction of *η* alone is better than that of Equation (2) with correction of *H_b_* alone. For each of the 3 Equations, there is no significant difference in the fitting errors obtained by different calculation levels.

All the fitted parameters of the 6 conditions are listed in Figure 1. Besides, to evaluate the effective boundaries of all the 6 conditions, the maximum absolute error *Δρ_max_* of each condition is calculated out and listed behind the fitted parameters.

As shown in Table 2, for the calculation level of B3PW91/6-31G(d,p), *Δρ_max_* calculated by Equations (1)–(3) are 0.15 g/cm^3^, 0.17 g/cm^3^ and 0.15 g/cm^3^, respectively, for the calculation level of B3LYP/6-31G**, they are 0.15 g/cm^3^, 0.16 g/cm^3^ and 0.15 g/cm^3^, respectively. The results of *Δρ_max_* are in accordance with the orders and comparisons demonstrated by Figure 2.

The results of Figure 2 and Table 2 indicate that the training accuracies caused by the ways to adopt correction items are ordered as *η + H_b_*, *η* and *H_b_*, and the 2 calculation levels B3PW91/6-31G(d,p) and B3LYP/6-31G** have the same accuracy in density prediction.

### 2.4. Evaluation of Accuracy and Generalization

To evaluate the accuracy and generalization of each Equation in Section 2.3, the 1847 compounds in the testing set are used to perform accuracy test. The former reported density prediction Equations and parameters, using above mentioned 2 calculation levels separately [20,21], are introduced as benchmarks to evaluate the accuracies of this work, as show in Figure 3.

As shown in Figure 3a that the predicted densities via Equation (12) using the parameters reported by Politzer et al. [21] are deviate from the experimental densities, especially when the densities are less than 1.6, and its EMS value is 37.8, a far greater value than that calculated by this work. As shown in Figure 3b–d that the predicted densities via the 3 Equations proposed by this work are close to the experimental densities, and the prediction MSEs via Equations (1)–(3) are 7.7, 8.0, and 7.2, respectively. For all 3 Equations, the MSE values are closer to that in Figure 2I–III, indicating that the prediction accuracy are perfectly in agreement with the training accuracy.

As shown in Figure 4a–d that the predicted densities, via Equation (12) using the parameters reported by Rice et al. [23] and the 3 Equations proposed by this work, are close to the experimental densities. The predicted MSEs via Equations (1)–(3) and (12) are 7.8, 8.0, 7.3 and 12.1, respectively, and the MSE values of Equations (1)–(3) are as good as that in Figure 2IV–VI. Figure 4 indicate that, for all the 3 Equations, the prediction accuracy is in good agreement with the training accuracy.

The effective boundaries of the above Equations and parameters to the testing set are evaluated via the maximum absolute error *Δρ_max_* of each condition in Figure 3 and Figure 4, and the values are listed in Table 3.

Table 3 shows that, for the QSPR of density prediction, *Δρ_max_* obtained by the parameters fitted based on Equations (1)–(3) are significantly smaller than that based on Equation (12) reported previously, whether the data set is calculated at B3PW91/6-31G(d,p) or B3LYP/6-31G**. Besides, for the three Equations proposed by this work, the values of *Δρ_max_* in Table 3 are very comparable with that in Table 2.

To judge the effective ratio (ER) of all the above mentioned Equations and parameters, the CDF profiles of *Δρ* are plotted using the testing set, as displayed in Figure 5 and Figure 6.

As shown in Figure 5, for *Δρ* < 5%, ER of Equations (1)–(3) and (12) are 92. 5%, 92.3%, 93.6%, and 52.4%, respectively. For *Δρ* < 2%, ER of Equations (1)–(3) and (12) are 53.4%, 51.0%, 53.6%, and 20.1%, respectively. It is clearly that all the ER values of (1)–(3) are much better than that of Equation (12) at the level of B3PW91/6-31G(d,p).

The same features are found in Figure 6, for *Δρ* < 5%, ER of Equations (1)–(3) and (12) are 92.3%, 91.4%, 92.9% and 85.4%, respectively. For *Δρ* < 2%, ER of Equations (1)–(3) and (12) are 53.7%, 51.4%, 53.2% and 45.4%, respectively. It is clearly that all the ER values of Equations (1)–(3) are better than that of Equation (12) at the level of B3LYP/6-31G**.

The results of this segment indicate that, for each of the Equations (1)–(3), the accuracy fitting from the training set can be perfectly reproduced by the testing set, and the accuracy of them are better than previous reported works fitted via Equation (12). It is verified that the accuracy of density prediction can be obviously improved by add molecular morphology *η* and hydrogen bond *H_b_* as correction terms in QSPR formulas. Equation (3) considering the coupling influences from molecular morphology and hydrogen bond is a reliable condition for density prediction of crystals. Both of the calculation level of B3PW91/6-31G(d,p) and B3LPY/6-31G** are capable of density prediction.

## 3. Theory and Method

### 3.1. Preparation of Data Set

The fundamental data set for training and testing used in this study were nitro compounds collected from the CCDC crystal database. The data set was built in four steps: (I) Crystal structure searching, (II) Data cleaning, (III) Molecular structure identification and (IV) DFT calculation.

Using –NO_2_ as substructure, more than 60,000 crystal structures were searched out from CCDC via step (I). Then the crystals containing elements outside the range of C, H, O and N or whose density values been measured under non-ambient conditions were removed in step (II). Step (III) extract individual molecules from each ‘cif’ file to identify and remove unreasonable compounds, including ionic, co-crystals, and molecules with H atoms missing. Finally, 3694 compounds have been kept after the first 3 steps. At step (IV), DFT calculation are performed at two calculation levels, B3PW91/6-31G(d,p) and B3LPY/6-31G**, in accordance with the previous reports [20,21], and both using D3BJ to preform surface analysis were applied using Multiwfn software [53,54] to get the degree of charge separation and CMED.

The training set was built by randomly picked out 50% of compounds from the data set, and the left 50% were used as testing set. The steps of (II), (III), and (IV) were performed on a high-throughput computing integration platform specially used for energetic materials, called Energetic Materials Studio1.0 (EMS 1.0) developed by Institute of Chemical Materials and Institute of Computer Applications. With the application of EMS 1.0, all the procedures of calculations and analysis were performed automatically.

### 3.2. Morphology Descriptor and Calculation Method

In a Cartesian coordinate system, any plane can be expressed as a universal Equation as
(4)Ax+By+Cz+D=0

The distance between an atom in a molecule and a given plane can be expressed as
(5)$$r_{i}=\frac{\left|x_{i}A+y_{i}B+z_{i}C+D\right|}{\sqrt{A^{2}+B^{2}+C^{2}}}$$
where (*x_i_*, *y_i_*, *z_i_*) is the coordinates of atom *i*, and *r_i_* is distance between atom *i* and the plane. The sum of *r_i_*^2^ of all atoms in the molecule is reasonable to evaluate the degree of coincidence between the shape of the molecule and the plane, which is expressed as
(6)$$\overset{N}{\underset{i=1}{\sum }}r_{i}^{2}=\sum \frac{{\left(x_{i}A+y_{i}B+z_{i}C+D\right)}^{2}}{\left(A^{2}+B^{2}+C^{2}\right)}$$

Suppose there is a plane in space to get the minimize value of $\sum_{i=1}^{N}r_{i}^{2}$, and the value $\min(\sum_{i=1}^{N}r_{i}^{2})$ can be used to measure how well the molecule shape matches a plane. Thus, a parameter to characterize the planarity of a molecule can be defined as
(7)$$p=\frac{\sqrt{min\sum_{i=1}^{N}r_{i}^{2}}}{N}$$
where *p* is planarity parameter, *N* is the total number of atoms in a molecule. It is implied by Equation (7), that small *p* vale means good planarity, and *p* = 0 means that all the atoms of the molecule are in the same plane. To solve the value of *p*, Particle Swarm Optimization algorithm [55] was applied, and the corresponding code is shown in the Appendix A.

In fact, it is not enough to use *p* as a single descriptor to evaluate planarity, and molecule size should be taken into account. As shown in Figure 7, among the molecules with similar *p*, large molecules are more plane-like than small ones.

Therefore, we introduced the distance between the farthest two atoms in a molecule to characterize molecule size
(8)$$r_{max}=\max\left(r_{ij}\right)$$
where *r_ij_* is the distance between any two atoms in a molecule, and *r_max_* is the distance between the farthest two atoms. Planarity and molecular size are considered simultaneously in the morphology of a molecule, by define a morphological descriptor *η* as
(9)η=e−2prmax
and the closer *η* gets to 1, the more planar the molecule is.

### 3.3. Hydrogen Bond Descriptor and Its Calculation Method

To quantitatively describe the hydrogen-bond environment of a molecule, a hydrogen-bond descriptor *H_b_* was introduced, which is concerned with the total number of hydrogen-bond donors of a molecule. In this study, we chose the H atoms connected with N atoms or O atoms as Hydrogen-bonded donors, and the hydrogen-bond descriptor *H_b_* is calculated via
(10)$$H_{b}=100\times \frac{\left(H_{N}+H_{O}\right)}{N}$$
where *H_N_* and *H_O_* are the H atoms connected to N atoms and O atoms, respectively, and *N* is the total number of atoms in the molecule. The corresponding code to calculate *H_b_* can be seen in Appendix A).

### 3.4. Functional Forms

The traditional formula [20,47] for predicting crystal density is
(11)$$\rho =\frac{M}{V_{m}}$$

The accuracy of density prediction is corrected and improved remarkably by introducing charge separation of positive and negative [21]
(12)$$\rho =\alpha \left(\frac{M}{V_{m}}\right)+\beta \left(\nu \sigma_{tot}^{2}\right)+\gamma$$
where *M* is the molecular mass, *V_m_* is the van der Waals volume of a molecule, $\sigma_{tot}^{2}$ is the variance of the total electrostatic potential on the molecular surface, ν is the charge balance degree. All the 3 descriptors are analyzed via Multiwfn software, where $\nu \sigma_{tot}^{2}$ is treated as one descriptor, and *V_m_* is acquire by calculate contour surface of electronic density with 0.001 electrons/bohr^3^ using an improved Marching Tetrahedron algorithm.

By evaluate the correlation between each descriptor and prediction errors, we introduce *η* and *H_b_* as new descriptors to correct function (12), and the new universal functional form for density prediction can be expressed as
(13)$$\rho =f\left(\frac{M}{V_{m}},\gamma \sigma_{tot}^{2},\eta ,H_{b}\right)$$

### 3.5. Accuracy Evaluatiom

Mean square error (MSE) analysis method is used for accuracy evaluation, and the relative percent error *Δρ* (%) between calculated density *ρ_cal_* and experimental density *ρ_exp_* is defined as
(14)$$\Delta \rho =100\times \frac{\left(\rho_{cal}-\rho_{exp}\right)}{\rho_{exp}}$$
(15)$$MSE=\frac{\sum_{i=1}^{N}\Delta \rho_{i}^{2}}{N}$$

To evaluate the generalization of all the QSPR functions, the cumulative distribution function (CDF) is applied by calculate the integral of the probability density function of *Δρ* via
(16)$$P\left(\Delta \rho \right)=\int_{0}^{\Delta \rho }\frac{N_{t}}{N}\times 100dt$$
where, *P*(*Δρ*) is the CDF of *Δρ*, *N* is the total number of molecules of the whole testing set, *N_t_* is the number of molecules at a given error *t*.

## 4. Conclusions

The influence of the two descriptors *η* and *H_b_* on prediction error indicated that Equations without considering these two factors will result in lower density prediction values, especially for molecules with more ‘bricklike’ shape or more hydrogen bonds. Adding *η* and *H_b_* as correction items in the conventional QSPR is expected to improve the accuracy of density prediction.

There are 6 working conditions were built considering the 3 QSPR models proposed by this work, and two calculation levels of B3PW91/6-31G(d,p) and B3LYP/6-31G**. The fitting result indicated that all the 3 Equations result in good accuracy with MSE < 9, and the calculation level of B3PW91/6-31G(d,p) is as good as B3LYP/6-31G**.

The accuracy and generalization of each Equation was verified by the 1847 compounds in the testing set. The accuracy of the 3 Equations fitting from the training set were perfectly reproduced by the testing set, and the accuracy and generalization of them are better than previous reported works fitted via Equation (12). At the level of B3PW91/6-31G(d,p), the ERs of the 3 Equations, for *Δρ* < 5%, are 92.5%, 92.3%, and 93.6%; for *Δρ* < 2%, the values are 53.4%, 51.0%, and 53.6%. At the level of B3LYP/6-31G**, for *Δρ* < 5%, the values are 92.3%, 91.4%, and 92.9%; for *Δρ* < 2%, the values are 53.7%, 51.4%, and 53.2%.

The accuracy of density prediction can be obviously improved by add molecular morphology *η* and hydrogen bond *H_b_* as correction terms in QSPR formulas. Considering both of *η* and *H_b_*, Equation (3) is a reliable condition for density prediction, and both B3PW91/6-31G(d,p) and B3LYP/6-31G** are applicable in density prediction.

## Figures and Tables

**Figure 1 molecules-25-00161-f001:**
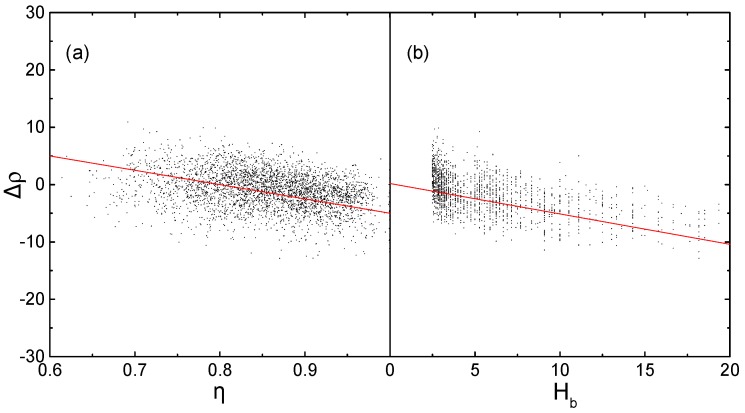
Relationships between the two descriptors and prediction errors, (**a**) *Δρ* versus *η*; and, (**b**) *Δρ* versus *H_b_*.

**Figure 2 molecules-25-00161-f002:**
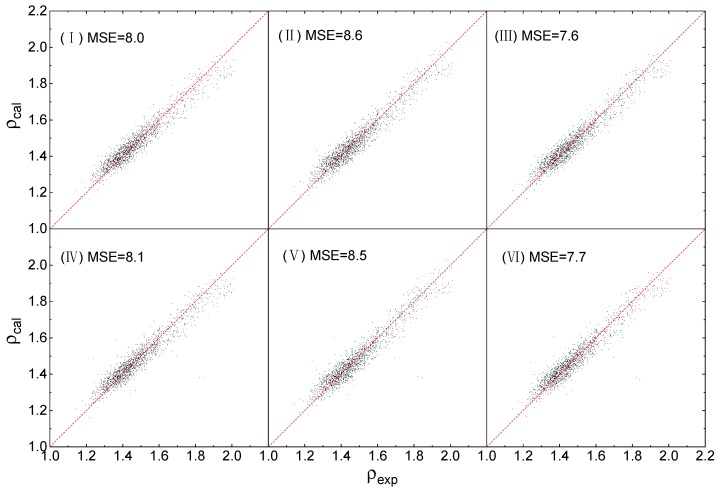
Calculated densities *ρ_cal_* versus experimental densities *ρ_exp_*.

**Figure 3 molecules-25-00161-f003:**
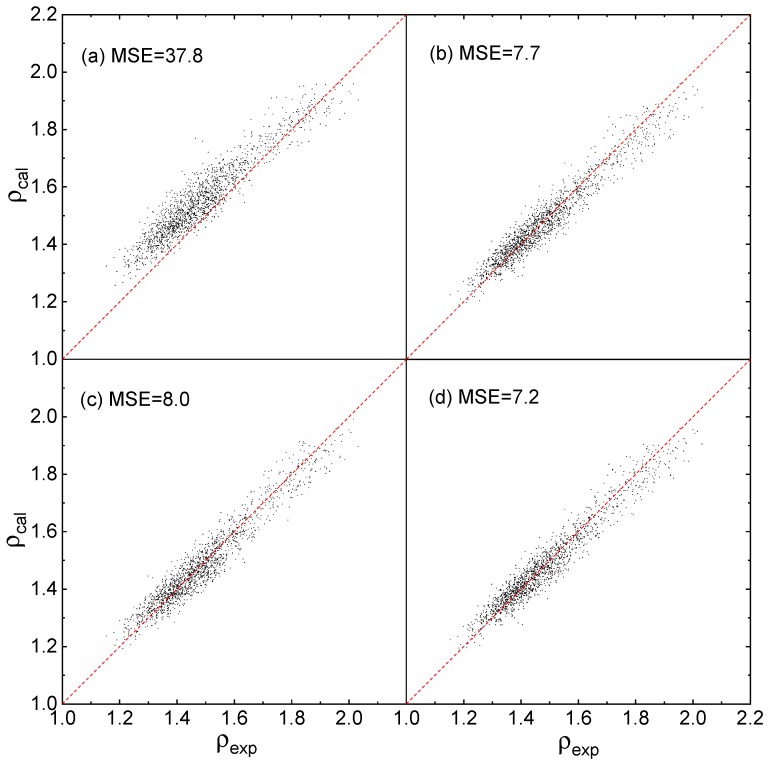
Predicted densities *ρ_cal_* versus experimental densities *ρ_exp_* at B3PW91/6-31G(d,p), (**a**) fitted by Politzer et al. [21] using Equation (12), (**b**–**d**) fitted by this work using Equations (1)–(3), respectively.

**Figure 4 molecules-25-00161-f004:**
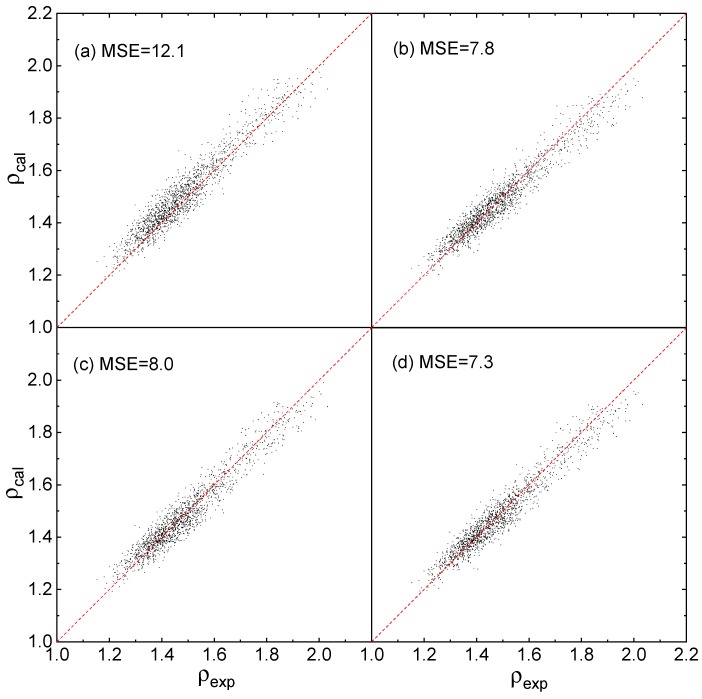
Predicted densities *ρ_cal_* versus experimental densities (ρ_exp_) at B3LYP/6-31G**, (**a**) fitted by Rice et al. [20] using Equation (12), (**b**–**d**) fitted by this work using Equations (1)–(3) respectively.

**Figure 5 molecules-25-00161-f005:**
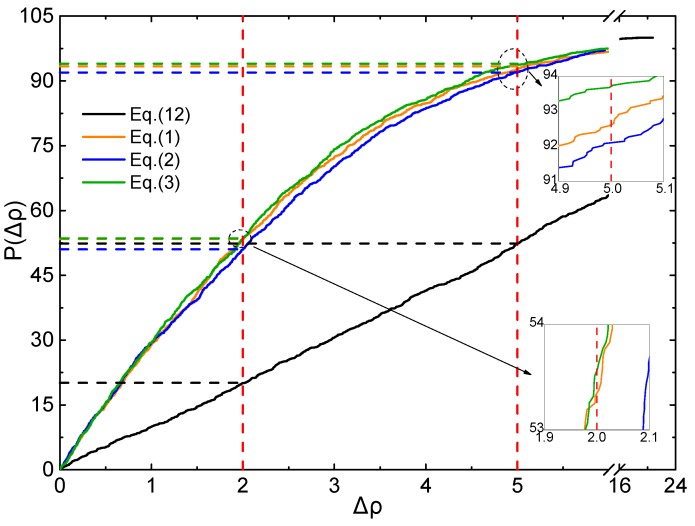
Profile of the CDF of *Δρ* for testing set (B3PW91/6-31G(d,p)).

**Figure 6 molecules-25-00161-f006:**
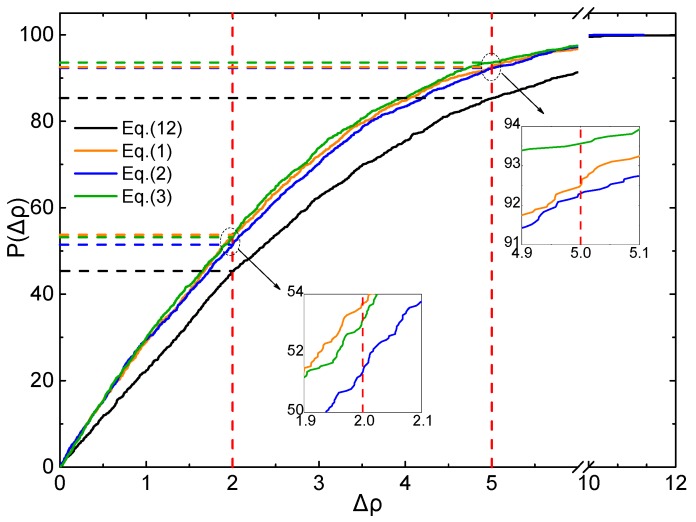
Profile of the CDF of *Δρ* for testing set (B3LYP/6-31G**).

**Figure 7 molecules-25-00161-f007:**
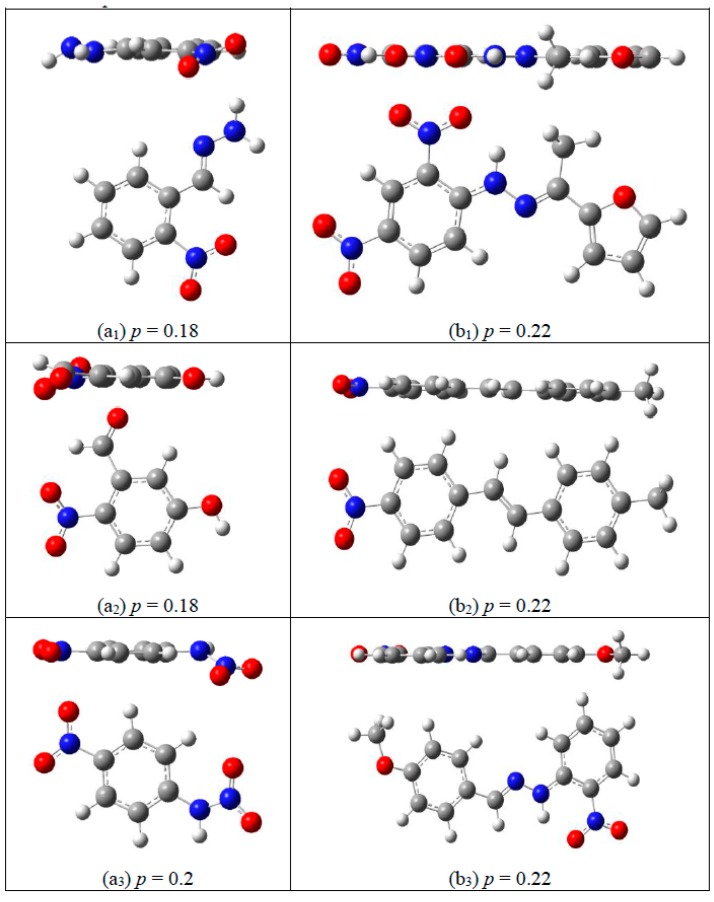
Planarity parameter *p* for smaller molecules (**a**) and bigger molecules; (**b**) displayed as side projections and front projections. The C, H, O, and N atoms are colored in gray, white, red, and blue, respectively.

**Table 1 molecules-25-00161-t001:** Working conditions for parameters fitting.

Condition	Equation	DFT Level
I	(1)	B3PW91/6-31G(d,p)
II	(2)	B3PW91/6-31G(d,p)
III	(3)	B3PW91/6-31G(d,p)
IV	(1)	B3LYP/6-31G**
V	(2)	B3LYP/6-31G**
VI	(3)	B3LYP/6-31G**

**Table 2 molecules-25-00161-t002:** Result parameters and maximum absolute error *Δρ_max_*.

Condition	α	β	β_1_	β_2_	γ	*Δρ_max_* (g/cm^3^)
I	1	0.0010	0.1960	-	−0.2549	0.15
II	1	0.0006	-	0.0036	−0.0825	0.17
III	1	0.0005	0.1836	0.0033	−0.2342	0.15
IV	1	0.0010	0.1802	-	−0.2320	0.15
V	1	0.0006	-	0.0036	−0.0721	0.16
VI	1	0.0005	0.1670	0.0033	−0.2105	0.15

**Table 3 molecules-25-00161-t003:** Maximum absolute error *Δρ_max_* for different calculation levels and fitting Equations.

Equation	B3PW91/6-31G(d,p)	B3LYP/6-31G**
(12)	0.33	0.21
(1)	0.16	0.15
(2)	0.15	0.15
(3)	0.15	0.15

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
