# Peer review of "Corrections of Molecular Morphology and Hydrogen Bond for Improved Crystal Density Prediction"

_molecules, 2019, doi:10.3390/molecules25010161_

Round 1
Reviewer 1 Report
Comments on Corrections of Molecular Morphology and Hydrogen Bond for Improved Crystal Density Predictions
The authors build on work expressing crystal density by a minimum number of parameters, which begins with a Group Additivity approximation in which a sum of fragment terms (atom or functional group) produces Volume and density estimates. [ref 20] The terms may be considered parameters and approximated by regression to experimental data. Closer connection with molecular properties can be established by using the computed molecular surface or (preferably) volume as represented by computed electronic isodensity surfaces. The adequacy of the additivity approximation can be judged by its performance for large and varied test sets. Issues arise in the choice of experimental or optimized individual molecular volumes. Optimized structures computed with B3LYP/6-31g(d,p) perform better for neutral (mean error in density 1%) than ionic systems. (mean error -5.1%). For ionic crystals the mean error in volume is reduced in magnitude by inclusion of a correction term for the number of H atoms, to -1.1%.
Improvement can also be achieved by including terms in the representation referring to the electrostatic potential on the computed isodensity surface, as proposed by Politzer [ref 21] and confirmed by Rice [ref 22]. The mean error in density estimates is negative, implying that the volume estimates are slightly too big.
The report under review uses a substantially larger data base than previous studies, and introduces two new descriptors. The first “eta” characterizes morphology, by combining a measure of planarity with molecular size. The second represents the effect of hydrogen bonding by the fraction of atoms which are H atoms bound to O or N. The error in predicted densities is linked to these two quantities, becoming more negative as eta approaches unity (the molecular planarity limit) and as HB grows (greater significance of hydrogen bonding). Each allows enhancement of density by improved packing (eta) or intermolecular attraction (HB).
Fitting including the two new terms is reported for (a) eta only; (b) HB only; and (c) both. All three cases a-c reduced the Mean Standard Error reported by Politzer (dramatically) and by Rice (modestly). Oddly, they all produced comparable MSE. This suggests a strong correlation among the terms.
This is a conscientious and successful effort to improve the estimates of crystal density by incorporating single-molecule terms which can serve as surrogates for intermolecular interactions. (The same can be said for the term referring to variance in electrostatic potential.) By its use of a much bigger data set than had been employed heretofore, the statistical weight is much enhanced.
I notice that in the fitting equation the coefficient alpha (multiplying the simplest M/V variable) is always greater than unity. This already corrects part of the error in the simplest formula. I see that alpha decreases as the new terms are included. It would give us a clearer picture of the significance of the new descriptors if either (1) the regressions were re-run with alpha set to unity, or (b) the correlation coefficient between parameters were evaluated.
The paper would benefit from copy-editing, there being numerous flaws in spelling and some in grammar. References to mathematical techniques (“Particle Swarm Optimization” and “Orthogonal Distance Regression”) would be welcome.
Author Response
Comments on Corrections of Molecular Morphology and Hydrogen Bond for Improved Crystal Density Predictions
The authors build on work expressing crystal density by a minimum number of parameters, which begins with a Group Additivity approximation in which a sum of fragment terms (atom or functional group) produces Volume and density estimates. [ref 20] The terms may be considered parameters and approximated by regression to experimental data. Closer connection with molecular properties can be established by using the computed molecular surface or (preferably) volume as represented by computed electronic isodensity surfaces. The adequacy of the additivity approximation can be judged by its performance for large and varied test sets. Issues arise in the choice of experimental or optimized individual molecular volumes. Optimized structures computed with B3LYP/6-31g(d,p) perform better for neutral (mean error in density 1%) than ionic systems. (mean error -5.1%). For ionic crystals the mean error in volume is reduced in magnitude by inclusion of a correction term for the number of H atoms, to -1.1%.
Improvement can also be achieved by including terms in the representation referring to the electrostatic potential on the computed isodensity surface, as proposed by Politzer [ref 21] and confirmed by Rice [ref 22]. The mean error in density estimates is negative, implying that the volume estimates are slightly too big.
The report under review uses a substantially larger data base than previous studies, and introduces two new descriptors. The first “eta” characterizes morphology, by combining a measure of planarity with molecular size. The second represents the effect of hydrogen bonding by the fraction of atoms which are H atoms bound to O or N. The error in predicted densities is linked to these two quantities, becoming more negative as eta approaches unity (the molecular planarity limit) and as HB grows (greater significance of hydrogen bonding). Each allows enhancement of density by improved packing (eta) or intermolecular attraction (HB).
Fitting including the two new terms is reported for (a) eta only; (b) HB only; and (c) both. All three cases a-c reduced the Mean Standard Error reported by Politzer (dramatically) and by Rice (modestly). Oddly, they all produced comparable MSE. This suggests a strong correlation among the terms.
This is a conscientious and successful effort to improve the estimates of crystal density by incorporating single-molecule terms which can serve as surrogates for intermolecular interactions. (The same can be said for the term referring to variance in electrostatic potential.) By its use of a much bigger data set than had been employed heretofore, the statistical weight is much enhanced.
Author Reply: Thank you for reading our articles carefully and sharing your valuable knowledge with us.
I notice that in the fitting equation the coefficient alpha (multiplying the simplest M/V variable) is always greater than unity. This already corrects part of the error in the simplest formula. I see that alpha decreases as the new terms are included. It would give us a clearer picture of the significance of the new descriptors if either (1) the regressions were re-run with alpha set to unity, or (b) the correlation coefficient between parameters were evaluated.
Author Reply: Thank you for your advice. The regressions were already re-run according to your advice, and the operations were pointed out as “The correction items, η and Hb have enough physical meanings, so the system correction factor α can be shielded, and all the values of α are set to 1 in the following fitting works.”, at line 205 and line 206. More importantly, all the related figures and tables have been replaced.
The paper would benefit from copy-editing, there being numerous flaws in spelling and some in grammar. References to mathematical techniques (“Particle Swarm Optimization” and “Orthogonal Distance Regression”) would be welcome.
Author Reply: Thank you for your advice. We asked a senior article author to polish the English writing, and the spelling and grammar were improved. References to Particle Swarm Optimization and Orthogonal Distance Regression were listed as Ref. 54 and Ref. 55, respectively.
Reviewer 2 Report
The authors have developed a model to develop better descriptors for molecular morphology and hydrogen bonding. The authors have taken a large data set to validate the model and they correctly identify the density trend. The study is extensive and well-articulated. They give a thorough description of the methods and it is easy to follow the steps.
However, there is a fundamental error in their choice of functionals. They have chosen two levels of DFT theory (B3PW91 and B3LYP). Since they are studying weak bonded interactions like H-bonding, they should have included other known functionals like D3/D2 or rVV10 or TS that includes weak bonded interaction energy in the molecules. These methods are widely studied and established. The authors should compare their model using either of the functionals or all of the functionals. Otherwise, the study is incomplete.
Author Response
The authors have developed a model to develop better descriptors for molecular morphology and hydrogen bonding. The authors have taken a large data set to validate the model and they correctly identify the density trend. The study is extensive and well-articulated. They give a thorough description of the methods and it is easy to follow the steps.
Author Reply: Thank you for your positive evaluation of our work.
However, there is a fundamental error in their choice of functionals. They have chosen two levels of DFT theory (B3PW91 and B3LYP). Since they are studying weak bonded interactions like H-bonding, they should have included other known functionals like D3/D2 or rVV10 or TS that includes weak bonded interaction energy in the molecules. These methods are widely studied and established. The authors should compare their model using either of the functionals or all of the functionals. Otherwise, the study is incomplete.
Author Reply: I totally agree with you that functional has influence on weak bonded interaction energy. According to your suggestion, we adopted d3bj in the two calculation levels to perform weak interaction corrections, which is no different from doing all the work over again. Thousands of molecules at 2 levels, all underwent optimization and frequency, and all the related figures and tables have been replaced. Most of the results are not significantly different from those without D3 correction done by the last version, except that the tow functionals used to fit density prediction QSPR are at the same accuracy level.
As you have learned from our article that our work is intended find a way to estimate crystal density by using single-molecule terms as surrogates for intermolecular interactions. Similar work has been reported by Politzer [Mol. Phys. 2009, 19, 2095-2101] and Rice [J. Phys. Chem. A. 2007, 111, 10874-10879] respectively. Therefore, the focus of this paper is to discuss the effectiveness of our new description compared with previous works, so that we chose the tow functionals just the same as the two former reports. I also agree that weak interactions between molecules require testing more functionals and correspondence correction methods, but this article deals only with the conformation of the molecules themselves and does not discuss intermolecular interactions.
Round 2
Reviewer 2 Report
Thank you for the work.